# 3-NAntC: A Potent Crotoxin B-Derived Peptide against the Triple-Negative MDA-MB-231 Breast Cancer Cell Line

**DOI:** 10.3390/molecules29071646

**Published:** 2024-04-06

**Authors:** Patricia Bezerra, Eduardo F. Motti

**Affiliations:** PHP Biotech International Inc., Keller, TX 76248, USA; patricia@phpbiotech.com.br

**Keywords:** cancer, triple-negative breast cancer, MDA-MB-231, Crotoxin, 3-NAntC

## Abstract

Breast cancer stands as the most prevalent type of tumor and a significant contributor to cancer-related deaths. Among its various subtypes, triple-negative breast cancer (TNBC) presents the worst prognosis due to its aggressive nature and the absence of effective treatments. Crotoxin, a protein found in the venom of *Crotalus* genus snakes, has demonstrated notable antitumor activity against aggressive solid tumors. However, its application has been hindered by substantial toxicity in humans. In efforts to address this challenge, Crotoxin B-derived peptides were synthesized and evaluated in vitro for their antitumor potential, leading to the discovery of 3-NAntC. Treatment with 3-NAntC at 1 µg/mL for 72 h notably reduced the viability of MDA-MB-231 cells to 49.0 ± 17.5% (*p* < 0.0001), while exhibiting minimal impact on the viability of HMEC cells (98.2 ± 13.8%) under the same conditions. Notably, 3-NAntC displayed superior antitumoral activity in vitro compared to cisplatin and exhibited a similar effect to doxorubicin. Further investigation revealed that 3-NAntC decreased the proliferation of MDA-MB-231 cells and induced G2/M phase arrest. It primarily prompted optimal cell death by apoptosis, with a lower incidence of the less desirable cell death by necrosis in comparison to doxorubicin. Additionally, 3-NAntC demonstrated low LDH release, and its cytotoxicity remained unaffected by the autophagy inhibitor 3-MA. In an in vivo zebrafish model, 3-NAntC exhibited excellent tolerability, showing no lethal effects and a low rate of malformations at high doses of up to 75 mg/mL. Overall, 3-NAntC emerges as a novel synthetic peptide with promising antitumor effects in vitro against TNBC cells and low toxicity in vivo.

## 1. Introduction

Data from the American Cancer Society estimate that in 2024, the USA might see around two million new cases of cancer and over 600,000 cancer-related deaths [1]. Breast cancer (BC) is globally recognized as the most prevalent tumor and the second leading cause of cancer-related mortality in the USA and Europe [2,3].

Triple-negative breast cancer (TNBC) accounts for 15–20% of all breast tissue tumors. TNBC cells are aggressive, invasive, and highly proliferative, with limited treatment alternatives, all contributing to the poorest prognosis among BC types [3]. TNBC cells lack expression of three receptors commonly found in BC: estrogen receptor (ER), progesterone receptor (PR), and the human epidermal growth factor receptor 2 (HER2). Therefore, TNBC cells cannot be targeted with drugs designed to modulate these receptors, making it the only BC subtype without specific targeted therapies [3,4].

Treatment strategies for TNBC typically include surgery, radiotherapy, and chemotherapy. Chemotherapeutic agents such as anthracyclines (e.g., doxorubicin), taxanes (e.g., paclitaxel), alkylating agents (e.g., cyclophosphamide), antimetabolites (e.g., capecitabine), and platinum-based drugs (e.g., cisplatin) may be employed in the neoadjuvant and/or adjuvant settings. Neoadjuvant chemotherapy, administered before surgery, is commonly recommended [3,4,5]. Recent advancements in TNBC treatment include the approval of novel drugs like Poly ADP Ribose Polymerase (PARP) inhibitors for patients with BRCA1/2 mutations, immune checkpoint inhibitors, and antibody-drug conjugates [3]. However, due to the severe side effects, toxicity, and limited efficacy of existing therapies, there remains an urgent need for new treatment modalities for TNBC.

Drug Research and Development based on natural proteins is a successful strategy. Poisons and venoms from animals are sources of proteins with a wide range of pharmacological activities. There are at least eleven approved drugs developed based on natural venoms and/or poisons, the majority from snake venoms, such as captopril, enalapril, batroxobin, and eptifibatide [6]. The first snake venom protein purified and crystallized was Crotoxin (CTX), the major venom component from *Crotalus durissus terrificus*, the South American rattlesnake [7]. Crotoxin’s antitumoral activity was demonstrated in vitro and/or in vivo in cell lines derived from breast (MCF-7) and lung (A549, SK-MES-1, and SPCA-1) aggressive human tumors [8,9,10,11]. Crotoxin reached the clinical development stage and underwent a phase I trial in patients with solid tumors refractory to conventional therapy. Crotoxin administration for 30 days reduced the tumor mass by >50% in 2 of 23 patients, and complete regression was observed in one patient; however, clinical trials were discontinued due to significant toxicity [12].

Crotoxin is a heterodimeric complex with a first basic, toxic phospholipase A2 (PLA2, Component B, CB, or crotoxin B) and a second acidic, non-toxic, non-enzymatic component (Component A or CA). The CA component increases the toxicity of the CB component. CA and CB components have isoforms with slight variations in enzymatic and pharmacological properties [7,13]. CTX complexes can be classified into two classes depending on their toxicity and enzymatic activity. CBb, CBc, and CBd isoforms (class I) complexed to CA are more toxic, have less phospholipase A2 activity, and dissociate from CA more slowly compared to class II CBa2 isoform (crotoxin CB1) [13].

The activities of Crotoxin (e.g., neurotoxicity, myotoxicity, nephrotoxicity, and cardiotoxicity) are associated with specific molecular regions [7]. However, the precise sequence responsible for toxicity remains unclear, although associations with the N-terminal, IBS (interfacial binding surface), or C-terminal regions have been proposed in independent studies [7,14,15]. Various modifications to Crotoxin have been explored in the literature to elucidate its pharmacological and toxic effects. Chemical modifications at four Crotoxin B amino acids and protein cleavage have altered its cellular lethality, myotoxicity, PLA2 enzymatic activity, bactericidal, edema-inducing, anticoagulant, and liposome-disrupting effects [16].

In this work, we produced different crotoxin B-derived peptides (CBa2 isoform) to characterize the regions responsible for the antitumoral and toxic activity. The strategy aimed to identify peptides that maintained the pharmacological effects with an improved safety profile.

## 2. Results

### 2.1. Synthesis and Characterization of Crotoxin B-Derived Peptides

The peptides discussed in the study were derived from fragments of the Crotoxin B isoform CBa2, chosen for its relatively lower toxicity compared to class I isoforms [13]. Figure 1 illustrates the amino acid sequence of Crotoxin B, highlighting residues that are crucial for crotoxin activity. These residues served as the basis for designing the peptides. Among these peptides, peptide 3 was referred to as 3-NAntC (Figure 1).

### 2.2. In Vitro Effects of the Crotoxin B-Derived Peptides in the Viability of Breast Cancer Cell Lines

Crotoxin B-derived peptides 1, 2, and 3 underwent evaluation in MDA-MB-231 (triple-negative breast cancer) and MCF10A (immortalized “normal” breast) cells to ascertain their antitumor and cytotoxicity profiles (Figure 2). Peptide 1 (2.5 µg/mL) did not decrease MDA-MB-231 cell viability at 24 and 48 h compared to the control treatment without peptides (94.4 ± 4.6% and 90.7 ± 3.0%, respectively, *p* > 0.05). Although peptide 2 (2.5 µg/mL) slightly decreased MDA-MB-231 viability at 24 (81.6 ± 5.0%, *p* < 0.05) and 48 h (85.0 ± 7.9%, *p* < 0.001), it also exhibited cytotoxicity against MCF10A benign cells to a similar extent after 48 h (85.7 ± 11.2%, *p* < 0.01). Therefore, peptides originating from the N-terminal regions were unable to selectively reduce the cell viability of breast cancer cells. Conversely, peptide 3 (0.8 µg/mL) significantly reduced the viability of MDA-MB-231 cells after 24 (77.5 ± 15.0%, *p* < 0.001) and 48 h (60.8 ± 17.9%, *p* < 0.0001) compared to non-treated cells. The viability was not significantly affected in MCF10A cells at 2.5 µg/mL (93.4 ± 7.4% and 80.5 ± 5.3% after 24 and 48 h, respectively, *p* > 0.05). However, higher concentrations of peptide 3 (20 µg/mL) reduced MCF10A cell viability after 48 h (65.9 ± 10.3%, *p* < 0.0001). These results indicate the specificity of peptide 3 activity.

Peptide 3, referred to herein as 3-NAntC, was resynthesized and characterized to confirm its activity. The resulting peptide exhibited high purity (>95%) and a molecular mass of 1645.95 g/mol (Appendix A). 3-NAntC was found in both monomeric and dimeric forms, with the abundance of the dimeric form increasing with higher pH levels (monomer/dimer ratio of 7.51 at pH 4.0, 6.47 at pH 7.0, and 27.31 at pH 10.0).

The activity of 3-NAntC on cell viability was then assessed in a broader panel of breast cancer (MDA-MB-231 and MCF-7), normal immortalized breast cells (MCF10A), human dermal fibroblasts (HDFa), and primary mammary epithelial (HMEC) cells (Figure 3). In benign cells, treatment with 3-NAntC at concentrations ≤ 1.0 µg/mL led to a slight decrease in cell viability (<20%). No significant differences in cytotoxicity were observed when comparing the treated MCF10A, HDFa, and HMEC cells to their respective untreated controls (*p* > 0.05).

On the other hand, 3-NAntC (1.0 µg/mL) significantly reduced the viability of MDA-MB-231 cells to 64.6 ± 14.5% (*p* < 0.05) and 49.0 ± 17.5% (*p* < 0.0001) after 48 and 72 h of treatment, respectively. Although 3-NAntC (1.0 µg/mL) also induced a decrease in the viability of MCF-7 cells, the effect was moderate and consistent even after prolonged exposure periods (54.7 ± 15.5% and 62.9 ± 26.0% after 48 and 72 h of treatment, respectively). The viability of MDA-MB-231 and MCF-7 cells was significantly lower than that observed for HMEC cells following 3-NAntC treatment (Figure 3d–f, *p* < 0.05). Moreover, triple-negative breast tumor cells (MDA-MB-231) were presented in lower numbers and have a greater number of vesicles compatible with apoptotic bodies when treated with 3-NAntC compared to no treatment (Appendix A).

Peptides 4–12, derived from 3-NAntC, were synthesized to explore the effects of amino acid modifications on their activity on cell viability (Appendix A). Peptides 5, 9, and 11 are cyclic due to intramolecular disulfide bonds between the N- and C-terminal cysteine residues. Despite exhibiting a significant decrease in cell viability in vitro for MDA-MB-231 cells at concentrations ≥ 0.4 μg/mL, the effects were inferior to those observed for 3-NAntC. (Appendix A). An additional in vitro experiment was conducted to evaluate if the activity of 3-NAntC could be replicated with a combination of peptides 6 and 7, which contain the same amino acids from 3-NAntC divided into two fragments. However, the combined effect of peptides 6 and 7 was lower than that observed for 3-NAntC. These findings further support that 3-NAntC is the smallest Crotoxin B-derived peptide maintaining potent inhibitory activity on the viability of TNBC cells.

The effect of 3-NAntC on MDA-MB-231 cells was compared to two chemotherapeutic drugs, doxorubicin and cisplatin (Figure 4). 3-NAntC significantly decreased MDA-MB-231 viability at concentrations ≥ 0.4 μg/mL at 24 and 48 h and ≥0.2 μg/mL at 72 h compared to the control (Figure 4a–c). However, the effect of 3-NAntC on HDFa cells was lower at doses ≤ 1.0 μg/mL, with cellular viability higher than 80% at all conditions (Figure 4d–f). 3-NAntC presented a higher inhibitory effect on MDA-MB-231 viability than cisplatin as early as 48 h of treatment, and its biological effect at 72 h was similar to that observed for doxorubicin at 48 h (Figure 4a–c). However, both 3-NAntC and cisplatin had a lower effect on the viability of HDFa cells compared to doxorubicin (Figure 4d–f).

### 2.3. Effect of 3-NAntC in Cell Proliferation and Cell Cycle Progression

A set of experiments was carried out to initially understand the mechanism of action associated with the activity of 3-NAntC on cell viability. The impact on cell proliferation was evaluated using the BrdU labeling method. 3-NAntC significantly reduced the proliferation rate of MDA-MB-231 cells at doses ≥ 0.2 μg/mL under all conditions (Figure 5a), with the maximum proliferation inhibition (71.1 ± 27.1%, *p* < 0.0001) observed at 0.6 µg/mL after 48 h of treatment. Interestingly, the reduction in proliferation was evident even at concentrations ≤ 0.2 µg/mL that did not affect cellular viability, indicating that this effect precedes cytotoxicity. Additionally, 3-NAntC did not alter the proliferation of HDFa cells at concentrations up to 1.0 µg/mL for 72-h exposures (Figure 5b).

A cell cycle progression assay was performed using PI labeling to quantify DNA content (Figure 5c). Since 3-NAntC did not affect the proliferation of HDFa cells, only MDA-MB-231 cells were analyzed. Following 48 h treatment, 3-NAntC at 0.8 µg/mL reduced the number of cells in the G0/G1 phase (−8.72 ± 4.03%, *p* < 0.01), while increasing cells in the G2/M phase (+8.08 ± 1.84%, *p* < 0.05). Thus, 3-NAntC impaired cells from progressing to the G0/G1 stage, while inducing G2/M phase arrest.

### 2.4. Mechanism of 3-NAntC-Induced Cellular Death

A flow cytometry analysis was performed using Annexin V and PI labeling was conducted to assess the effects of 3-NAntC on the apoptosis/necrosis cellular profile (Figure 6a–c). 3-NAntC primarily induced apoptosis in tumor cells and reduced necrosis compared to doxorubicin. At 24 h, 3-NAntC exhibited a significantly lower percentage of necrotic cells compared to doxorubicin. Moreover, a significant increase in the apoptosis rate was observed at 24 and 48 h after treatment with 3-NAntC at 0.8 µg/mL (+20.9 ± 11.1%, *p* < 0.05 and +12.4 ± 3.8%, *p* < 0.001, respectively), while maintaining the percentage of cells in necrosis. By 72 h, 3-NAntC increased the apoptosis rate at all tested doses, reaching a maximum rate of 36.9 ± 6.6% (*p* < 0.0001) at 0.8 µg/mL.

Furthermore, the LDH release assay was conducted to determine the effects of 3-NAntC on damaging the cellular plasma membrane. The results indicated that 3-NAntC treatment did not consistently increase LDH release from MDA-MB-231 cells for the 24 and 48 h treatments (Figure 6d), suggesting that the peptide does not induce cell death by necrosis or pyroptosis under these conditions [17]. Although at 72 h there was an increase in LDH release, this result could be attributed to the absence of macrophages to engulf the post-apoptotic cells [18].

### 2.5. Cell Viability in the Presence of the Autophagy Inhibitor 3-MA

A cell viability assay was conducted in the presence and absence of 3-MA to determine the effects of 3-NAntC on autophagy induction. The results revealed that 3-MA was unable to reverse the decrease in MDA-MB-231 viability caused by treatment with 3-NAntC (Appendix A). Therefore, these findings suggest that 3-NAntC did not induce cell death through autophagy mechanisms.

### 2.6. In Vivo Tolerability in Zebrafish

In vivo experiments using zebrafish embryos were performed to evaluate the toxicity of 3-NAntC. Low doses of 3-NAntC (1.5 to 150 µg/mL) and a higher dose (75 mg/mL) did not show lethality or significant non-lethal events (malformations) (Appendix A). However, at the dose of 150 mg/mL, 3-NAntC caused 45% lethality after 48 and 72 h of treatment, with 50–60% of the surviving embryos exhibiting malformations (Appendix A). Despite these effects, the LD50 (the dose that kills 50% of the embryos) could not be determined even with extremely high doses of 3-NAntC, demonstrating the remarkable tolerability of zebrafish to this peptide.

## 3. Discussion

Crotoxin is a potent heterodimeric neurotoxin with promising activity against tumoral cells, albeit with limited applicability in the clinic due to marked toxicity in humans [19]. In a phase I clinical trial, Crotoxin was administered intramuscularly to patients with solid tumors that were resistant to conventional therapies. Neuromuscular toxicity, including diplopia, strabismus, nystagmus, and eyelid ptosis, were the main adverse events observed in 18 of 23 patients [12,20]. These results corroborated the local myotoxicity and muscle necrosis previously noted in rats [21] and systemic skeletal muscle damage observed in mice [22]. Considering the toxicity found in recent clinical trials using Crotoxin, researchers are trying to mitigate the side effects by using intravenous administration, as well as applying a gradual dose escalation [20].

The molecular mechanism and specific amino acid residues responsible for Crotoxin’s pharmacological activities, including its antitumoral and neurotoxic effects, have not been fully elucidated [13,23]. Identifying these residues consists of a promising strategy to develop new Crotoxin-based peptides with reduced myotoxicity, nephrotoxicity, and cardiotoxicity [13]. Venom-based substances have already resulted in approved drugs for several diseases, such as exenatide for Type 2 diabetes, ziconotide for chronic pain, lepirudin and desirudin for thromboembolic disease, and eptifibatide for acute coronary syndrome [24]. In oncology, synthetic peptides derived from venoms have reached the clinical development stage, including the chlorotoxin-derived peptide 131I-TM601 for the treatment of gliomas and the soricidin-derived peptide SOR-C13 for solid tumors overexpressing the TRPV6 ion channel [25,26]. Synthetic peptides based on PLA2 that are present in *Bothrops brazili*, *Bothrops jararacussu*, *Bothrops asper*, and *Agkistrodon piscivorus piscivorus* snake venoms were shown to exhibit cytotoxic activity against human cancer cell lines [27,28,29]. For Crotoxin B, although no previous structure–antitumoral activity relationship was described, previous studies showed that its cleavage resulted in a significantly reduced myotoxicity and PLA2 activity, while only partially affecting antibacterial activity and in vivo lethality in mice [16].

In the present work, Crotoxin B-derived peptides were shown to impair the viability of the breast cancer cell lines MCF-7 and MDA-MB-231, with a lower negative effect against benign cell lines. Although Crotoxin activity in breast cancer cells had been previously demonstrated against MCF-7 and ER+ aromatase-overexpressing breast cancer (MCF-7aro) cells [10,19], this is the first time that this antitumoral activity is being explored for novel therapies targeting TNBC. 3-NAntC, a peptide originating from the C-terminal region of Crotoxin B, was able to selectively reduce the viability of breast cancer cells, unlike the peptides from the N-terminal region. This result is aligned with the antitumoral activity observed for peptides derived from PLA2 C-terminal region from venoms from other snake species, such as *Bothrops brazili* and *Bothrops jararacussu* [27,28,29].

3-NAntC decreased the proliferation rate of tumor cells, thereby impairing the tumor growth, invasion, and metastasis [30]. This mechanism of action complements the impairment of the cell viability, as any surviving cells may be prevented from multiplying. In MDA-MB-231 cells, 3-NAntC induced cell cycle arrest at the G2/M phase, causing cells to accumulate before the final step of division and becoming more susceptible to apoptosis [31]. Indeed, a higher rate of apoptosis was observed in MDA-MB-231 cells treated with 3-NAntC, as confirmed by the flow cytometry assay using Annexin V and PI labeling. Comparatively, Crotoxin was shown to induce cell cycle arrest at the G2/M phase in MCF-7aro cells [19], while G0/G1 and S phase arrest was observed in lung (SPCA-1) and esophageal (Eca-109) tumors [9,32].

To understand the mechanism of action of 3-NAntC, the peptide-induced autophagy was evaluated, as this process can be associated with treatment-induced cellular death or with drug resistance [33]. However, cell viability following 3-NAntC treatment remained unaffected by the presence of the autophagy inhibitor 3-MA, thus indicating that the peptide does not induce this cellular process in MDA-MB-231 cell lines. This result also suggests a lower potential for the 3-NAntC treatment to induce drug resistance via this cellular process. Crotoxin was shown to induce autophagy in MCF-7 and SK-MES-1 cells, but not in MCF-7aro, suggesting that different mechanisms are involved in the pharmacological response depending on the cell line and its intrinsic features [10,19]. Moreover, Crotoxin-induced autophagy in MCF-7 cells may be associated with cell membrane disruption, since pretreatment with the autophagy inhibitor 3-MA was previously shown to reduce LDH release [10].

A comparison of toxicity with the chemotherapeutic drug doxorubicin revealed a higher occurrence of necrosis in MDA-MB-231 cells treated with doxorubicin than with 3-NAntC. This finding is consistent with the low LDH release observed following 3-NAntC treatment compared to the literature data for doxorubicin [34].

The 3-NAntC peptide has demonstrated promising effects in the current research; however, the authors acknowledge that several aspects remain to be elucidated. Future directions of this research aim to delve into not only the physical–chemical properties of the peptide, including its plasma stability, but also to gain a deeper understanding of its mechanism of action. Moreover, there is a need to explore how this peptide may exert its effects in other subtypes of triple-negative breast cancer (TNBC) and various types of tumors beyond TNBC. This comprehensive approach will not only enhance our understanding of the therapeutic potential of the 3-NAntC peptide but also pave the way for its broader application in cancer treatment.

## 4. Methods

### 4.1. Synthesis and Characterization of Crotoxin-Derived Peptides

Peptides were synthesized by WatsonBio (Houston, TX, USA) through amino acids coupling and high-efficiency, solid-phase synthesis assisted by microwaves. The peptides were then purified using high-efficiency liquid chromatography coupled with a UV-visible detector (HPLC-UV-vis). Briefly, purification was conducted within a semi-preparatory scale (5 mg of sample and flux of 1 mL/min) at room temperature. This process was performed on a Varian Pro Star 210 chromatograph (Varian Medical Systems, Palo Alto, CA, USA) and with a Pro Star 330 UV detector (Varian Medical Systems, Palo Alto, CA, USA). The columns used were 4.6 × 250 mm and Boston Green 0DS-AQ was employed as the stationary phase (loops of 5 μL, and 250 μL, respectively). A gradient of phases A (0.1% trifluoroacetic in 100% water) and B (0.1% trifluoroacetic in 100% acetonitrile) was employed as follows: 0.01 min (88:22, respectively), 25 min (63:37, respectively), and 25.1 min (0:100), stopping at 30 min. The flow rate was 1.0 mL/min, the injection volume was 10 μL, and detection was at 220 nm. For sequence confirmation, MS/MS followed the HPLC and was conducted as follows: Probe: ESI, Probe bias: 4.5 kV, Nebulizer Gas Floe: 1.5 L/min, Detector: 1.5 kV, CDL: −20.0V, CDL Temp: 250 °C, Block temperature: 200 °C, T. Flow: 0.2 mL/min and B. conc: 50%H_2_O/50%ACN. Peptide sequences were determined based on MS/MS fragmentation spectra using the Andromeda search engine and MaxQuant environment (Max-Planck-Institute of Biochemistry, Planegg, Germany) [35]. Peptides 1 and 2 were obtained based on the N-terminal region of Crotoxin B (NH2 to COOH, HLLQFNKMIKFETRKNAIPF and AIPFYAFY, respectively), while Peptide 3 was obtained based on the C-terminal region (MFYPDSRCRGPSET). Nine other peptides were produced from fragments and/or the cyclization of peptide number 3.

### 4.2. Cell Culture

MCF-7 (Luminal breast cancer), MDA-MB-231 (TNBC), and MCF10A (non-tumor mammary gland epithelial cells) were purchased from the Rio de Janeiro Cell Bank (BCRJ, Rio de Janeiro, RJ, Brazil). HMEC (Primary Mammary Epithelial Cells) and HDFa (Human Dermal Fibroblasts) cell lines were purchased from Gibco (Waltham, MA, USA). MCF-7 and MDA-MB-231 were cultured in an RPMI 1640 medium supplemented with 10% heat-inactivated FBS, 1% glutamine, and 1% antibiotic/antimycotic solution. MCF10A and HMEC were cultured in MEBM medium supplemented with 100 ng/mL cholera toxin, 2.5 mM L-glutamine, 5% horse serum, 10 µg/mL human insulin, 0.5 µg/mL hydrocortisone, and 10 ng/mL EGF. All cells used in this study were incubated at 37 °C under 5% CO_2_ and checked for authenticity through polymerase chain reaction (PCR) followed by fragment analysis of eight highly polymorphic microsatellite loci (short tandem repeat—STR) plus gender determination with score ≥ 80%.

Stock solutions of Peptides 1 to 12, doxorubicin, and cisplatin were prepared in PBS (1X). Doxorubicin and cisplatin were stored at −20 °C and the peptide solutions were freshly prepared before each experiment. The final concentration of PBS (1X) in the culture medium was lower than 0.02%. All the negative controls contained this vehicle in the same culturing conditions as previously described.

### 4.3. Cell Viability Assay

Cell viability assay was conducted using the MTT method [36]. For treatments lasting 24 h, the cells were seeded at the density of 2 × 10^4^ cells/well into 96-well sterile plates, and for treatments lasting 48 or 72 h, the cell density was 1 × 10^4^ cells/well. The following day, cells were treated by applying 200 μL/well of each agent (Peptides 1 to 12, doxorubicin, or cisplatin) at the desired concentration and incubated for the duration of the treatment. Afterward, cells were incubated with 20 μL/well of an MTT solution (5 mg/mL) for 2.5 h. Then, formazan salts were solubilized with 200 μL of DMSO: isopropanol (3:1) and agitated for at least 15 min at room temperature. The absorbance was measured at 570 nm with a spectrophotometer (Molecular devices Spectra Max Plus 384—San Jose, CA, USA), and relative cell viability was calculated. To evaluate the mechanism of cell death, cell cultures were treated with peptides in the presence or absence of 3-methyladenine (3-MA) (1nM), which is an autophagy inhibitor [37]. After treatment, the MTT assay was carried out as previously described.

### 4.4. Cell Proliferation Assay Using the BrdU Method

Cell proliferation was assessed with the BrdU method [38] using the Cell Proliferation ELISA, BrdU (colorimetric) kit from Roche (Basel, Switzerland). Cell seeding and treatments were carried out as presented in Section 4.3. BrdU-labeling (20 μL/well) was added and incubated for 2.5 h. Then, ELISA was performed with an anti-BrdU-POD antibody, followed by washing and substrate solutions. The substrate was then developed for 15 min and the reaction was stopped with HCl (6M). The absorbance was measured at 495 nm with a spectrophotometer (Molecular devices Spectra Max Plus 384—San Jose, CA, USA), and the relative cell proliferation was calculated.

### 4.5. Cell Cycle Progression Assay

A cell cycle progression assay using flow cytometry and propidium iodide (PI) was conducted to determine the effects of 3-NAntC on the different cell cycle phases [39]. For this assay, cells were seeded into a 6-well sterile plate at a density of 2.5 × 10^5^ cells/well. The following day, cells were treated with 3-NAntC with doses varying from 0.2 to 0.8 µg/mL for 48 h. Afterward, cells were collected, washed with 1mL of PBS, and fixed with cold 70% ethanol. Then, cells were washed with PBS, resuspended with 135 µL of RnaseA solution at 100 µg/mL, and incubated at room temperature for 30 min. Cells were labeled with 2 µL PI at 100 µg/mL and analyzed for DNA content with a flow cytometer (BD FACsCANTO II—San Jose, CA, USA), based on the acquisition of 100,000 events. A linear scale was set for the detectors of the fluorescence channels and for forward (FSC) and side (SSC) light scatter. Debris, cell doublets, and aggregates were gated out using a two-parameter plot of FL-2-Area to FL-2-Width of PI fluorescence. Data were analyzed using the BD FACsDiva™ 8.0.3 (Franklin Lakes, NJ, USA) and ModFit LT 5.0 (Verity Software House, Topsham, ME, USA) analysis software. The antiproliferative effects were indicated by the percentage of cells in the G0/G1, S, and G2/M phases of the cell cycle.

### 4.6. Cellular Death Assessment

Labeling with the fluorescent dyes Annexin V-FITC and PI enables the discrimination of cellular death by apoptosis and necrosis [40]. Cells were seeded into a 6-well sterile plate at a density of 2.5 × 10^5^ cells/well. The following day, cells were treated with 3-NAntC (doses varying from 0.4 to 0.8 µg/mL) or doxorubicin (2 µg/mL) for the duration of the treatment. Afterward, cells were collected and washed with 200 μL of a binding buffer (1×) from the FITC Annexin V Apoptosis Detection Kit I (BD Biosciences, Franklin Lakes, NJ, USA). Cells were resuspended in 100 µL/tube of binding buffer (1×) and 1 µL of Annexin V-FITC was added to each tube except the blank. Cells were then incubated for 20 min on ice and in the dark. Two µL of PI (100 µg/mL) was added right before the read with the flow cytometer (BD FACsCANTO II—San Jose, CA, USA), based on the acquisition of 100,000 events. Debris, cell doublets, and aggregates were gated out using a two-parameter plot of FL-2-Area to FL-2-Width of PI fluorescence. Then, a logarithmic scale was set for the detectors of the fluorescence channels and a bivariant analysis of Annexin V and PI distinguished different cell subpopulations: Annexin V^−^/PI^−^ were designated as viable cells; Annexin V^+^/PI^−^ and Annexin V^+^/PI^+^ as apoptotic and late apoptotic, and Annexin V^−^/PI^+^ as necrotic cells.

### 4.7. Cellular Death Assay Using the LDH Release Method

The CyQUANT LDH Cytotoxicity Assay (Thermo Fisher, Waltham, MA, USA) was employed to determine LDH release as a measure of plasma membrane damage, an important indicator of necrosis and pyroptosis [41]. It is known that LDH extravasation does not occur in apoptosis due to the formation of apoptotic bodies and engulfment of material by leukocytes [17,42]. Cell seeding and treatments proceeded as presented in Section 4.3. After treatment incubation, 20 μL of supernatant was transferred to an EIA/RIA plate. The Reaction Mixture (20 μL/well) was added and incubated at room temperature for 30 min in the dark. Afterward, 60 μL/well of the Stop Solution was added and the absorbance of the reaction mixtures was quantified at both 490 nm and 680 nm with a spectrophotometer (Molecular devices Spectra Max Plus 384—San Jose, CA, USA). The absorbance at 680 nm was subtracted from the value at 490 nm, and then relative LDH release was calculated.

### 4.8. In Vivo Zebrafish Toxicity Study

Transgenic Tg(*fli*1:EGFP)^y1^ zebrafish embryos were raised at 28 °C for 48 h in E3 embryo medium 0.2 mM 1-Phenyl-2-Thiourea (PTU). Unfertilized eggs or larvae that did not appear healthy or exhibited any obvious developmental defects were excluded before treatment onset (~10%). Zebrafish embryos obtained at 48 h post fertilization (hpf), n = 20 embryos/group, were subcutaneously injected with the 3-NAntC peptide at 0.0015, 0.015, 0.15, 75, and 150 mg/mL diluted in E3 embryo medium containing 0.003% 1-phenyl-2-thiourea (PTU) and the embryos were maintained at 28.5 °C. Pictures were taken after 24, 48, and 72 h of peptide exposure and the percentage of live animals, as well as the percentage of live animals with malformations, were visually analyzed.

### 4.9. Statement of Ethical Approval

The animal experiments were conducted following institutional ethical guidelines. The protocols were approved by the Institutional Animal Care and Use Committee (IACUC) of BioReperia AB (Linköping, Sweden). Data reporting follows the recommendations of the ARRIVE guidelines.

### 4.10. Statistical Analysis

The in vitro cell viability, cell proliferation, cell cycle progression, and cell apoptosis/necrosis profile were analyzed using the two-way ANOVA followed by the Dunnett or Bonferroni post hoc test to the significance of 5%. LDH release was analyzed using the one-way ANOVA followed by the Dunnett post hoc test to the significance of 5%. The normality score for each data set used in the present study is presented in the Appendix A. GraphPad Prism v6.0 was used for the analysis.

## 5. Conclusions

In summary, we have described the discovery and characterization of 3-NAntC as a novel active peptide against the TNBC cell line MDA-MB-231. 3-NAntC demonstrates efficacy in impairing the growth and survival of tumor cells while having minimal to no effects on non-tumor cells. It blocks cell cycle progression by inducing G2/M phase arrest and primarily promotes cell death via apoptosis, with lower levels of necrosis than conventional therapies and no evident involvement of autophagy.

This work represents the first description of a synthetic peptide derived from Crotoxin component B (CBa2 isoform) with reduced toxicity and enhanced antitumoral activity against TNBC cells. 3-NAntC emerges as a promising drug candidate for TNBC treatment, to be further investigated in additional in vitro and in vivo preclinical studies.

## Figures and Tables

**Figure 1 molecules-29-01646-f001:**
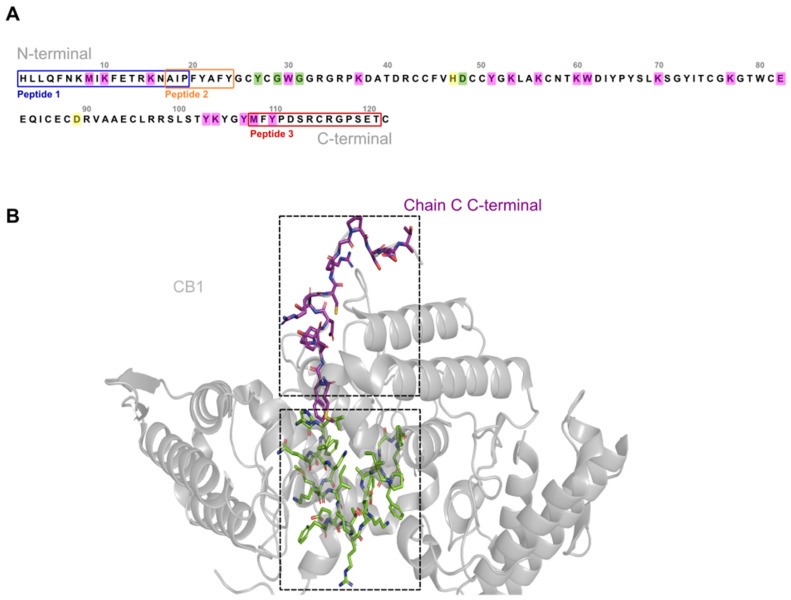
**Amino acid sequence and structure of Crotoxin B (PLA_2_) from *Crotalus durissus terrificus* snake venom**. (**A**) Amino acid sequence of Crotoxin B. The regions that originated the peptides 1, 2, and 3 are shown within the blue, orange, and red boxes, respectively. The residues from the catalytic site are highlighted in yellow, while the residues from the Ca^2+^ binding site are highlighted in green. The residues highlighted in purple are important for Crotoxin B toxicity/lethality, myotoxicity, edema induction, antibacterial, liposome-disrupting, enzymatic, and anticoagulant activities. (**B**) Structure of Crotoxin B (PDB: 2QOG) with the regions that originated peptides 1 and 2 highlighted with carbon atoms in green and peptide 3 highlighted with carbon atoms in purple.

**Figure 2 molecules-29-01646-f002:**
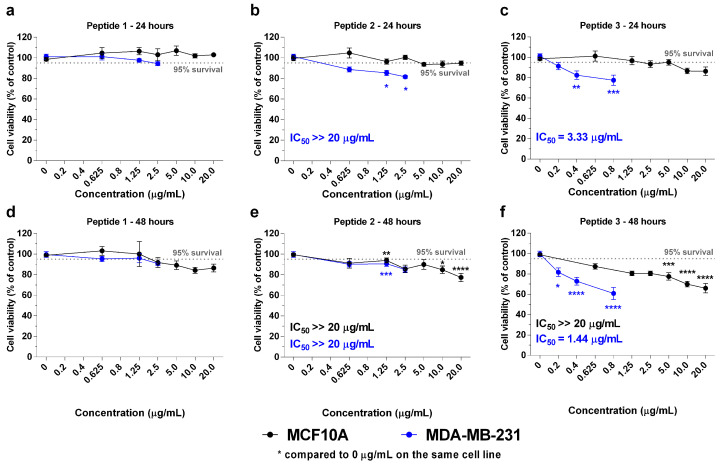
**Evaluation of Crotoxin B-derived peptides on cellular viability against TNBC cells and mammary benign cells**. MDA-MB-231 (blue line) and MCF10A cells (black line) were treated with peptides 1 (**a**,**d**), 2 (**b**,**e**), and 3 (**c**,**f**) for 24 and 48 h at the concentration range of 0.2–20 µg/mL. Cellular viability assay was evaluated using the MTT method. Data are shown as mean ± SEM of at least three independent assays. Significant differences between control (0 µg/mL) and treated cells are designated as * *p* < 0.05, ** *p* < 0.01, *** *p* < 0.001, and **** *p* < 0.0001, according to two-way ANOVA and a Bonferroni post hoc test.

**Figure 3 molecules-29-01646-f003:**
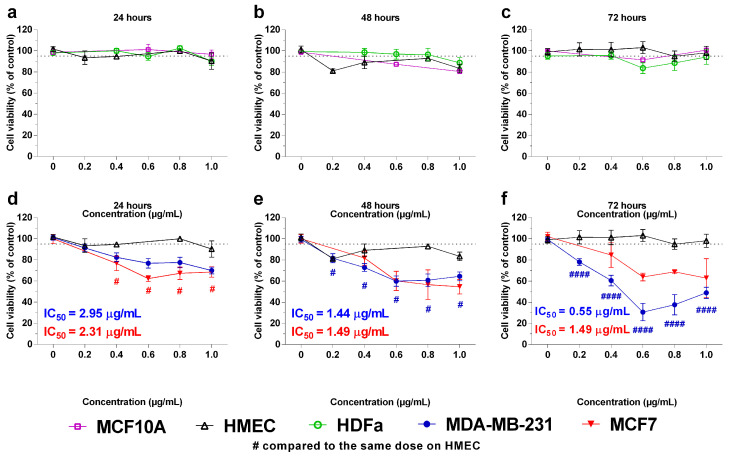
**Activity of 3-NAntC (peptide 3) in breast cancer cell lines compared to benign cells**. Tumor (MDA-MB-231 and MCF-7) and benign (HDFa, HMEC, and MCF10A) cell lines were treated with 3-NAntC for 24, 48, and 72 h at the concentration range of 0.2–1.0 µg/mL. Cellular viability assay was conducted using the MTT method. Data are shown as mean ± SEM of at least three independent assays in triplicate. (**a**–**c**) Benign cell lines treated with the same concentrations of 3-NAntC were compared to the control (0 µg/mL) using two-way ANOVA and a Bonferroni post hoc test. (**d**–**f**) MDA-MB-231 and MCF-7 tumor cell lines were treated with various concentrations of 3-NAntC and compared to the HMEC benign cell line using two-way ANOVA and a Bonferroni post hoc test, # *p* < 0.05, and #### *p* < 0.0001.

**Figure 4 molecules-29-01646-f004:**
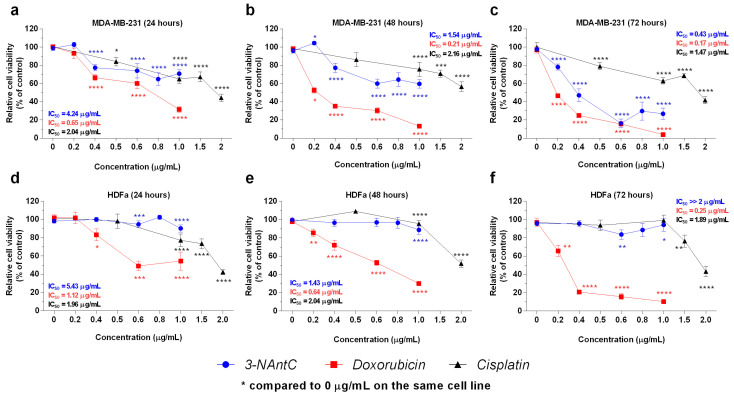
**Comparison of cell viability between 3-NAntC and chemotherapy (doxorubicin and cisplatin).** Cellular viability assays were performed on MDA-MB-231 (**a**–**c**) and HDFa (**d**–**f**) cell lines using the MTT method. Cells were treated with 3-NAntC, cisplatin, or doxorubicin for 24, 48, and 72 h at concentrations ranging from 0.2 to 2.0 µg/mL. Data are presented as mean ± SEM of at least three independent assays in triplicate. Significant differences between the control (0 µg/mL) and treated cells are designated as * *p* < 0.05, ** *p* < 0.01, *** *p* < 0.001, and **** *p* < 0.0001, according to two-way ANOVA and a Bonferroni post hoc test.

**Figure 5 molecules-29-01646-f005:**
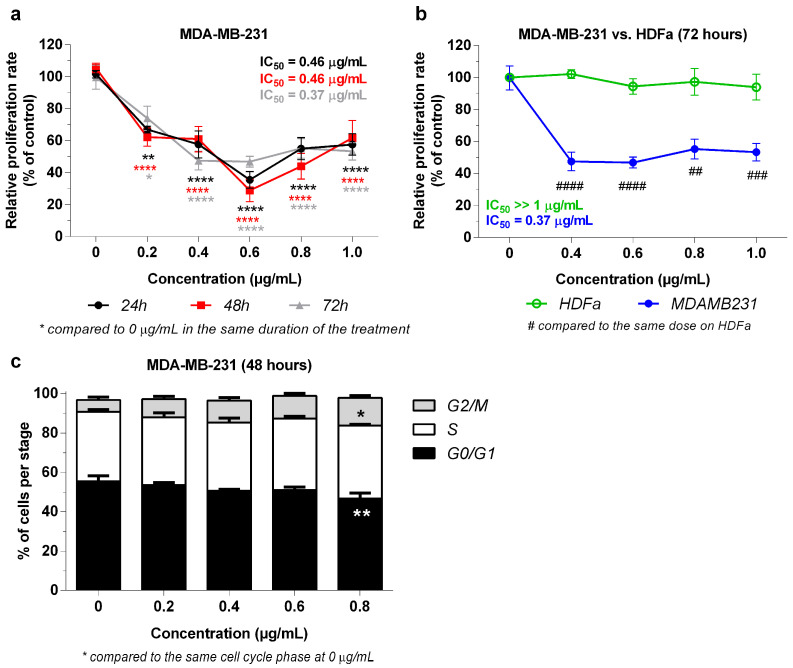
**Antiproliferative effects of 3-NAntC in TNBC and normal cells**. (**a**,**b**) Cellular proliferation assay using the BrdU incorporation method. MDA-MB-231 and HDFa cells were treated with 3-NAntC for 24, 48, and 72 h at a concentration range of 0.2–1.0 µg/mL. Data are shown as a mean ± SEM of at least three independent assays in triplicate. * *p* < 0.05, ** *p* < 0.01, and **** *p* < 0.0001 show significance compared to the control (0 µg/mL), and ## *p* < 0.01, ### *p* < 0.001, and #### *p* < 0.0001 show significance compared to the same concentration in HDFa cell lines according to the two-way ANOVA and a Bonferroni post hoc test. (**c**) Cell cycle progression assay. Cells were treated with 3-NAntC for 48 h at the concentration range of 0.2–0.8 µg/mL. A flow cytometry assay was conducted with propidium iodide labeling. Data are shown as a mean ± SEM of at least three independent assays. Significant differences between the control (0 µg/mL) and treated cells are designated as * *p* < 0.05, and ** *p* < 0.01, according to the two-way ANOVA and Dunnett post hoc test.

**Figure 6 molecules-29-01646-f006:**
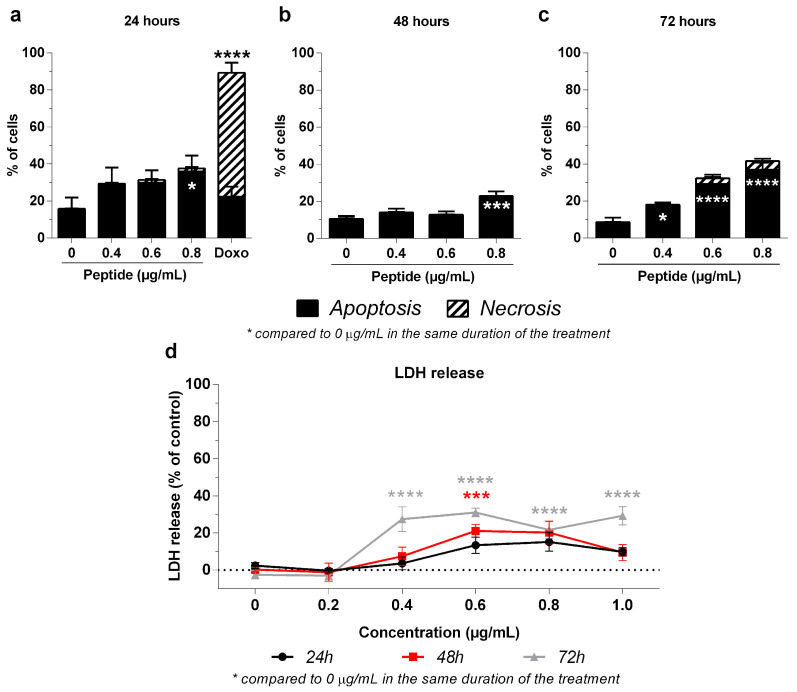
**Effect of 3-NAntC in apoptosis/necrosis profile of the MDA-MB-231 cell line**. (**a**–**c**) A flow cytometry with Annexin V and Propidium iodide labeling was performed. Cells were treated with the peptide 3-NAntC (0–0.8 µg/mL) or with doxorubicin (0.2 µg/mL) for 24, 48, and 72 h. 3-NAntC induced cell death by apoptosis with a very low amount of necrosis. Data are shown as a mean ± SEM of at least three independent assays. (**d**) LDH release assay. Cells were treated with the 3-NAntC peptide (0–1.0 µg/mL) for 24, 48, and 72 h and the LDH release was quantified. There is no significant difference in this biomarker, suggesting no involvement of either necrosis or pyroptosis as a primary cell death mechanism. Data are shown as mean ± SEM of at least three independent assays in triplicate. Significant differences between control (0 µg/mL) and treated cells are designated as * *p* < 0.05, *** *p* < 0.001, and **** *p* < 0.0001, according to the two-way ANOVA and Dunnett post hoc test.

## Data Availability

The original contributions presented in the study are included in the article/Appendix A, further inquiries can be directed to the corresponding author.

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
