# Peer review of "3-NAntC: A Potent Crotoxin B-Derived Peptide against the Triple-Negative MDA-MB-231 Breast Cancer Cell Line"

_molecules, 2024, doi:10.3390/molecules29071646_

Round 1
Reviewer 1 Report
Comments and Suggestions for Authors
The study focuses on developing a synthetic peptide, 3-NAntC, derived from Crotoxin B, to target triple-negative breast cancer (TNBC). Researchers synthesized and evaluated 3-NAntC in vitro, demonstrating its significant reduction in TNBC cell viability while sparing healthy cells. Compared to traditional chemotherapeutic agents, 3-NAntC showed superior efficacy and induced apoptosis with minimal toxicity. In vivo testing in a zebrafish model confirmed its excellent tolerability. Overall, 3-NAntC emerges as a promising novel peptide with potent antitumor effects against TNBC and low toxicity in preclinical models.
Comments:
1. It would be beneficial to include some discussion on the limitations of the study and potential future directions for research in this area.
2. The results should be validated in at least one more c=TNBC cell line, preferable, the widely used MDA-MB-468.
3. Can the authors highlight the key structural and functional difference that makes this modified peptide, less toxic and more effective than previous peptide that went in the clinical trial?
4. In section 2.2 and others, please provide IC50 data and calculation for each cell line used. It’s difficult to understand why control cells have data upto20ug/ml while the study cells data is only up to 0.8 or 2.5ug/ml.
5. What was the solvent used for dissolving 3-NAntC and what concertation max can be achieved. Does this compound have good solubility? If the compound was dissolved in anything apart from water, did the authors use the relevant solvent on control cells? If so, what was the percentage?
6. In all the graphs, please clearly mention what comparison groups were used to calculate the p-values? Just looking at the p-values, it is difficult to interpret the comparison groups in graphs with multiple cell line data.
7. The methods section needs extensive revision. There are incomplete sentences, wrong units and difficult to read sentences. Please modify it as extensively as possible with neat details and correct language. (Example: ethanol 70. (line 416), conic (line 414, 427), completing the volume to (line 385) etc.)
8. For cellular death assessment assay, it would be good if the authors provide the gating strategy and results for each cell line as flow graphs followed by quantification graphs.
9. The overall presentation of the figures, colors can be improved.
10. Please provide the normality score for each dataset used in two-way ANOVA.
11. What re the future directions for this peptide? Can the authors validate the toxicity of this compound in mouse? This data will be highly supportive of the findings if it is feasible for the authors to generate TNBC mouse model and treat with the compound. This comment is a recommendation and subjective to the fact if the authors have the facility for animal studies.
Comments on the Quality of English LanguageNeeds significant improvement. There are a few areas where improvements could enhance clarity. For instance, some sentences could be streamlined for smoother flow, and there are a few instances where punctuation or word choice could be optimized for better readability.
Author Response
We appreciate the input you provided for our work. Please review the comments for each question below:
1. We have added a new paragraph at the end of the discussion section:
"The 3-NAntC peptide has demonstrated promising effects in the current research; however, the authors acknowledge that several aspects remain to be elucidated. Future directions of this research aim to delve into not only the physical-chemical properties of the peptide, including its plasma stability but also to gain a deeper understanding of its mechanism of action. Moreover, there is a need to explore how this peptide may exert its effects in other subtypes of triple-negative breast cancer (TNBC) and various types of tumors beyond TNBC. This comprehensive approach will not only enhance our understanding of the therapeutic potential of the 3-NAntC peptide but also pave the way for its broader application in cancer treatment."
2. We recognize that validation is crucial for our research, and we intend to conduct it in the near future, not only with other types of TNBC but also with other types of tumors. In the present study, we aimed to establish a point of reference for future comparisons, allowing us to ascertain whether this peptide exhibits enhanced efficacy in particular tumor types. Furthermore, this research has laid the foundation necessary for advancing to in vivo studies in rodents, enabling a comprehensive assessment of the therapeutic potential of 3-NAntC.
3. The peptide under investigation possesses several characteristics that render it active with reduced toxicity. Of particular note is its absence of sequences from the active enzymatic site of Crotoxin B, a phospholipase A with the potential to interact with cell membranes and induce pore formation, leading to necrosis. Another notable feature is its size, which is approximately 10 times smaller than Crotoxin B, and its net charge. These properties enable the peptide to approach the cell membrane more effectively and interact specifically with receptors, particularly in the RCR region. This information has not been published yet and will be included in the future MoA research paper.
4. We have included the IC50 values for all curves that exhibited a significant decrease in cell viability or cell proliferation. For benign cell lines, we maintained the higher doses on the graph to demonstrate that the peptide maintains safety in vitro in these models even at higher concentrations.
5. 3-NAntC was solubilized in PBS at pH 7.4. Solubility experiments revealed that this peptide exhibits high solubility in PBS 100 mM, with an estimated peptide solubility at 24ºC of 136 mg/mL. We have included a new paragraph in the Materials and Methods, section 5.2, to clarify this matter:
"Stock solutions of Peptides 1 to 12, Doxorubicin, and Cisplatin were prepared in PBS (1x). Doxorubicin and Cisplatin were stored at -20ºC and peptide solutions were freshly prepared before each experiment. The final concentration of PBS (1x) in the culture medium was lower than 0.02%. All the negative controls contained this vehicle in the same culturing conditions as previously described."
6. We have added this information to each graph.
7. The whole text was extensively reviewed for English. Specially, in the Material and Methods section, a lot of information was added.
8. The gating strategy was added to the Materials and Methods, sections 5.5 and 5.6. While we acknowledge the visual appeal of flow graphs, we believe that given the complexity of our data (four treatments over three durations, plus controls), integrating them would clutter the figure without significantly enhancing its informational value. Therefore, we've decided to retain the figure in its original form for clarity and coherence.
9. The color panel was improved.
10. Please see answer no. 1.
Reviewer 2 Report
Comments and Suggestions for Authors
The manuscript characterizes different peptides derived from Crotoxin to identify the protein regions responsible for its antitumoral and toxic activities. The aim is to find peptides that retain the pharmacological effects of Crotoxin while improving safety profiles, thus potentially offering new therapeutic options for TNBC treatment.
Please consider addressing following comments-
1. Why the peptide is referred as 3-NAntC.
2. Given the previous toxicity concerns associated with Crotoxin, the manuscript should include experiments to assess the toxicity of the synthesized peptides. This could involve testing the peptides on healthy cells or animal models to evaluate potential adverse effects such as neurotoxicity, myotoxicity, nephrotoxicity, or cardiotoxicity.
3. Does author consider performing Structure–Activity Relationship Analysis by synthesizing and testing different peptide variants (3-NAntC) with modifications at specific amino acid residues. This could help identify critical regions within the peptides responsible for their pharmacological effects and toxicity.
4. Can author comment on half-life of synthesized peptide.
5. Micrography picture of MCF-7 or MDA-MB-231cells treated with peptide 0-0.8 μg/ to monitor cell morphology would enhance the finding.
Comments on the Quality of English Language
The manuscript is well written.
Author Response
We appreciate the input you provided for our work. Please review the comments for each question below:
- 3-NantC was chosen as a preliminary commercial name to facilitate the reference to peptide 3. It stands for Triple (3) Negative (N) anti (ant) cancer (C).
- Preliminary toxicity was assessed through the in vitro assays by using three types of benign cells: MCF10a (an immortalized breast-originated cell line), HMEC (a primary breast-originated cell line), and HDFa (a primary epithelial-originated cell line). Mammary cell lines were used to assess the local toxicity of these new peptides, while the epithelial cell line was used to assess the extended toxicity throughout the body. No obvious toxicity was noted in vitro for the doses that impaired the tumor cells' viability and proliferation. Moreover, the Zebrafish model, an FDA-approved model for toxicity, was used to assess the preliminary toxicity in vivo (please see the Supplementary material), showing great tolerability even for doses way higher than the in vitro IC50. It is worth mentioning that the LD50 was not reached by these high doses, and malformations, cardiotoxicity, neurotoxicity, and necrosis were not observed up to 75 mg/mL of 3-NAntC.
- An initial structure-activity relationship analysis was shown in the Supplementary Material, where we demonstrated that 3-NANtC is the minimum amino acid sequence needed for the anti-tumor activity. Also, we demonstrated that cyclization would impair this activity and that Cysteine in position 8 is crucial for the whole antitumor potential. Our recent findings showed that both size and net charge may also be important for its biological activity. These properties enable the peptide to approach the cell membrane more effectively and interact specifically with receptors, particularly in the "RCR" amino acids' region. This information has not been published yet and will be included in the future MoA research paper.
- Half-life studies were not yet completed, however, we noted that for in vitro studies we lost activity after 15 days in solution. So, we decided to solubilize all peptides freshly for each assay. We have also made a plasma stability for 3-NAntC and findings showed that this peptide would not last even 5 minutes in plasma. Since the beginning, we understood that the peptide would not be our final formulation for the in vivo studies, so we have developed a novel platform to carry this peptide and provide the chemical and thermal stability we desired. This information has not yet been published and will be included in a future paper describing the proof-of-concept in rodents.
- The micrography picture of MDA-MB-231 cells untreated or treated with 3-NAntC at 1 µg/mL was added to the Supplementary material.
Round 2
Reviewer 1 Report
Comments and Suggestions for Authors
The authors have meticulously responded to all the comments and the manuscript is suitable for acceptance. Thank you.